# Clinical and Genetic Re-Evaluation of Inherited Retinal Degeneration Pedigrees following Initial Negative Findings on Panel-Based Next Generation Sequencing

**DOI:** 10.3390/ijms23020995

**Published:** 2022-01-17

**Authors:** Kirk A. J. Stephenson, Julia Zhu, Adrian Dockery, Laura Whelan, Tomás Burke, Jacqueline Turner, James J. O’Byrne, G. Jane Farrar, David J. Keegan

**Affiliations:** 1Mater Clinical Ophthalmic Genetics Unit, The Mater Misericordiae University Hospital, D07 R2WY Dublin, Ireland; tomasburke@mater.ie (T.B.); jturner@mater.ie (J.T.); jamesobyrne@mater.ie (J.J.O.); dkeegan@mater.ie (D.J.K.); 2Next Generation Sequencing Laboratory, Pathology Department, The Mater Misericordiae University Hospital, D07 R2WY Dublin, Ireland; AdrianDockery@mater.ie; 3The School of Genetics & Microbiology, Trinity College Dublin, D02 PN40 Dublin, Ireland; whelan11@tcd.ie (L.W.); jane.farrar@tcd.ie (G.J.F.)

**Keywords:** inherited retinal degenerations, retinal dystrophy, genetic testing, next generation sequencing, whole exome sequencing, single gene sequencing, unresolved inherited retinal degenerations

## Abstract

Although rare, inherited retinal degenerations (IRDs) are the most common reason for blind registration in the working age population. They are highly genetically heterogeneous (>300 known genetic loci), and confirmation of a molecular diagnosis is a prerequisite for many therapeutic clinical trials and approved treatments. First-tier genetic testing of IRDs with panel-based next-generation sequencing (pNGS) has a diagnostic yield of ≈70–80%, leaving the remaining more challenging cases to be resolved by second-tier testing methods. This study describes the phenotypic reassessment of patients with a negative result from first-tier pNGS and the rationale, outcomes, and cost of second-tier genetic testing approaches. Removing non-IRD cases from consideration and utilizing case-appropriate second-tier genetic testing techniques, we genetically resolved 56% of previously unresolved pedigrees, bringing the overall resolve rate to 92% (388/423). At present, pNGS remains the most cost-effective first-tier approach for the molecular assessment of diverse IRD populations Second-tier genetic testing should be guided by clinical (i.e., reassessment, multimodal imaging, electrophysiology), and genetic (i.e., single alleles in autosomal recessive disease) indications to achieve a genetic diagnosis in the most cost-effective manner.

## 1. Introduction

Inherited retinal degenerations (IRDs) are rare genetic disorders associated with pathogenic variation in over 300 known genetic loci, manifesting with variably progressive visual dysfunction [1]. IRDs represent the main cause of visual loss in the working age group in many Western nations [2,3]. The clinically and genetically heterogeneous nature of IRDs make accurate molecular diagnosis challenging. First-tier genetic testing of IRDs with panel-based next-generation sequencing (pNGS) has a diagnostic yield of ≈70–80% [4,5,6]. Whole exome/genome sequencing (WES/WGS) may detect further variants, resolving up to 79% of pedigrees [7,8]; however, this increased scope of sequencing comes with increased cost and requires resources to process and store data as well as effectively manage significant secondary findings [9,10].

Meticulous phenotyping may guide the choice of most appropriate genetic testing modality [11]. Novel therapeutic options including gene and stem cell therapies are burgeoning, and confirmation of genetic etiology is often a prerequisite for access to clinical trials and approved treatments [12,13,14,15]. Confirmation of genotype is important also for family risk determination so that families with genetically determined IRDs can be evaluated, tested, and counseled appropriately with regard to reproductive options (e.g., prenatal diagnosis, pre-implantation genetic testing). Thus, it is critical to maximise the genetic resolve rate for IRDs.

Herein, we describe the process of reassessing the phenotype of patients with an initial negative genetic result from pNGS and the rationale of further clinical care and genetic testing strategies.

## 2. Methods

Patients enrolled on the Mater Clinical Ophthalmic Genetics arm of the Irish national inherited retinal degeneration registry (Target 5000) were assessed for a genetic cause of their ophthalmic ± syndromic phenotype. All patients had undergone comprehensive clinical assessment including LogMAR visual acuity (VA, Optos plc, Dunfermline, Scotland, UK), formal visual fields (VF, Humphrey Field Analyser, Carl Zeiss MediTec, Dublin, CA, USA), colour vision assessment (Lanthony D15, Gulden Ophthalmics, Elkins Park, PA, USA), ocular motility/nystagmus assessment, and dilated slit lamp biomicroscopy with Goldmann applanation tonometry (Haag-Streit UK Ltd, Harlow, UK). Multimodal imaging included colour fundus photography and autofluorescence (Optos ‘California,’ Optos plc, Dunfermline, Scotland, UK) and spectral domain optical coherence tomography (OCT, Cirrus 5000, Carl Zeiss MediTec, Dublin, CA, USA). Visual electrophysiology (ERG, Metrovision, Perenchies, France) was assessed where diagnostically relevant. All clinically assessed patients had undergone a research grade pNGS of 250 IRD-implicated genes at the Ocular Genetics Unit, Trinity College Dublin [4,5,16].

All patients with a negative result from an initial pNGS approach (exonic regions of 250 IRD-implicated genes) were reassessed by reviewing their existing records (clinical examinations, multimodal imaging, visual fields, and electrophysiology). This was performed by three clinicians in a masked fashion (KS, TB, DK). A fourth investigator (JZ) assessed these reports for agreement, with consensus cases progressing to the clinical genetics multidisciplinary team (MDT) and disputed cases brought back for in-person clinical reassessment. In-person reassessment included functional (VA, VF, electrophysiology) and structural assessment (multimodal imaging) as required for diagnostic clarification. See Figure 1A.

Unresolved pedigrees that retained a phenotype consistent with IRD were referred to the clinical genetics MDT for discussion of the most appropriate further genetic testing modalities based on their clinical findings, family history, and any findings from 1st-tier pNGS (e.g., single allele in autosomal recessive disease). The second-tier genetic testing modalities employed in this study are as follows. For cases in which gene coverage from the initial pNGS run was felt to be inadequate, a repeat pNGS sequencing (250 gene panel) run and/or direct manual inspection of Binary Alignment Map (BAM) files was performed (pedigrees 19, 20, 23). For cases where primary pNGS coverage of genes of interest was adequate, a larger pNGS of 351 genes was applied to extracted DNA by a commercial laboratory (Blueprint Genetics, Helsinki, Finland) (pedigrees 24, 26) [17]. Gene panels can be found in Appendix A. Single gene testing was applied to cases with a single candidate variant previously detected on 1 allele or to cases with a classic phenotype e.g., Stargardt Disease (OMIM#248200). This approach included sequencing of both exons and introns of the gene of interest and flanking regions, in particular *ABCA4* (OMIM*601691, pedigrees 21 and 22), *ADGRV1* (OMIM*602851, pedigree 31), *BBS1* (OMIM*209901, pedigree 32), *CDH23* (OMIM*601067, pedigree 29), *CNNM4* (OMIM*607805, pedigree 28), *EYS* (OMIM*612424, pedigree 25), *PEX7* (OMIM*601757, pedigree 18), and *TRIM32* (OMIM*602290, pedigree 30). Trio WES (sequencing of all exonic DNA and flanking regions) was performed by a commercial laboratory (Blueprint Genetics) for 4 pedigrees (proband and their parents) where no candidate variants had been identified on 1st-tier pNGS. Three pedigrees undergoing trio WES had a non-syndromic retinitis pigmentosa (RP) phenotype (one each had X-linked, autosomal dominant, and autosomal recessive modes of inheritance, pedigrees 27, 33, and 34, respectively) with the remaining pedigree having an autosomal dominant vitreoretinopathy phenotype (pedigree 35). All detected genetic variants were reported with HGNC nomenclature, confirmed with bidirectional Sanger sequencing (to rule out incorrect reads, etc.), and compared with the reference genome (GRCH37/HG19) (see Figure 1B).

## 3. Results

Of 441 patients (331 pedigrees) in the Mater Clinical Ophthalmic Genetics Unit IRD cohort that underwent genetic testing before 2019, 69 patients (16%) from 52 pedigrees were genetically unresolved after 1st-tier pNGS. Following reassessment of their phenotypes, 74% of patients (*n* = 51) retained a clinical diagnosis of IRD, and 26% (*n* = 18) were deemed non-IRD acquired disease (Table 1 and Figure 2). Mean age was 58.06 (SD ± 16.97) years and 50.57 (SD ± 16.12) years, and 61% and 53% were female for the acquired and IRD groups, respectively. Of the 51 patients (35 pedigrees) who remained clinically consistent with IRD, 34 patients (67%) from 18 pedigrees were available for further genetic testing (Table 2). A total of 17 clinically IRD patients were unavailable for further investigation (deceased (*n* = 2), no family member available for trio WES (*n* = 4), patient preference due to SARS-CoV-2 pandemic (*n* = 11)).

Clinical reassessment has allowed a revision of the clinical diagnosis (i.e., non-IRD) in 18 patients and, with the genetic investigation techniques outlined in Table 2, a further 16 patients from 10 IRD pedigrees have been resolved bringing the total genetic resolution rate for IRDs in this cohort to 92% (388/423).

Repeated pNGS testing led to genetic resolution in 100% of patients assessed in this manner (*n* = 5, pedigrees 19, 20, 23, 24, 26). If coverage with the 1st-tier pNGS (250 gene panel) approach was adequate, the larger 351 gene panel was used for 2nd-tier genetic assessment; however, if coverage of genes deemed to be relevant to the clinical phenotype was inadequate (e.g., *BBS10*), repeat of the initial 250 gene panel was performed.

Single gene testing (introns and exons) was used for cases of clinically autosomal recessive IRDs where one pathogenic allele had been previously identified on pNGS. This approach was used for nine cases, with 44% detecting a second pathogenic allele (pedigrees 18, 21, 22, and 25).

A total of 15 patients (four pedigrees) with no candidate genetic variants from 1st-tier pNGS underwent WES, but three out of four pedigrees remain unresolved (pedigrees 34, 35, and 36). In total, four novel variants in *ABCA4*, *EYS*, *FLVCR1*, and *RPGR* were detected (Table 2). Pedigrees that remain unresolved following 2nd-tier testing will undergo array comparative genomic hybridisation and/or WGS techniques to assess for structural variants or copy number variations.

The 18 patients (17 pedigrees) deemed non-IRD on clinical re-evaluation have all been referred to the appropriate ophthalmic sub-specialties (e.g., uveitis, neuro-ophthalmology, medical retina) and subsequently discharged from the IRD clinic. 

## 4. Discussion

For 69 patients (16%) previously unresolved by 1st-tier pNGS, clinical reassessment guided the most appropriate course of action: further genetic testing modalities in 74% and reclassification as acquired disease in 26%. Nearly half (47%) of patients (16 of 34) or 56% of pedigrees (10 of 18) available for 2nd-tier genetic reassessment were genetically resolved after further investigation, including four novel variants (Table 2). Following these additional tests, 92% (388/423) of this total IRD cohort received a molecular diagnosis for their retinal/syndromic phenotype.

### 4.1. Clinical Reassessment

Useful factors for confirmation of IRD were (1) symptom onset before 40 years, (2) symmetrical disease, (3) positive family history, (4) signs of progression, and (5) associated ocular/systemic features (e.g., juvenile onset posterior subcapsular cataract, sensorineural hearing loss, post-axial polydactyly, etc.) [18,19]. Typical retinal phenotypes suggestive of IRD can be seen in Figure 3. Conversely, non-IRD diagnoses were associated with unilateral or asymmetrical disease (e.g., uveitis, trauma, Figure 2) and onset after 60 years (e.g., age-related macular degeneration). These factors are not individually/independently diagnostic of IRD, but considering the weight of the total evidence can be suggestive. Many acquired retinal conditions with advanced manifestations may mimic IRDs with features such as arteriolar attenuation (e.g., retinal vasculitis, retinal arteriolar occlusion), disc pallor (e.g., anterior ischemic optic neuropathy, glaucoma) and intraretinal RPE migration (e.g., retinal pigment epitheliitis, late-stage multifocal choroiditis) [20]. In this cohort, significant interocular asymmetry was found in 0% (0/16) of resolved IRD cases versus 33.33% (6/18) of non-inherited cases. Thus, this finding should prompt thorough clinical and biochemical analysis for an acquired cause before considering IRD genotyping. Subjective (e.g., past medical history, family history, cadence of progression) and objective (e.g., clinical aspect, multimodal imaging, electrophysiology) clues help to clarify the likelihood of a genetic etiology for each case of retinal degeneration. Classic mimickers of IRD include autoimmune retinopathy, infectious and non-infections posterior uveitides, and drug-induced retinal toxicity (e.g., hydroxychloroquine (simulating bullseye maculopathy/STGD1), deferoxamine). In the initial stages of this research collaboration, difficult cases with some IRD features were progressed to genetic testing in the hope of useful insights from genetic data. This approach has now changed with the benefit of MDT experience showing that the addition of potentially unrelated genetic findings may obfuscate the true etiology, adding unnecessary delay and anxiety for the affected patient and their wider family. A negative result from appropriate genetic screening for IRDs (e.g., pNGS) should prompt clinical reassessment for an acquired cause. This may enhance the percentage of resolved pedigrees in all cohorts as reasonable exclusion of a primary genetic etiology has been performed with a minimum of overlooked pedigrees (i.e., acquired cases are removed from the total number in cohort).

### 4.2. Second-Tier Genetic Testing Approaches and Costs

The panel-based NGS approach in a research-based academic laboratory with validation in an accredited laboratory adopted by Target 5000, as outlined in previous publications, accrued substantial cost savings (Table 3) [5,16]. Using pNGS compared to WES as a first-pass approach for ‘gene hunting’ is favorable, as it reduces bioinformatic demand, identifies the molecular diagnosis for the majority of IRD patients, and frees resources for 2nd-tier testing of more difficult cases as necessary (Figure 1B). Similar approaches have been adopted by other centers with resolution of approximately one-third of partially resolved cases on 2nd-tier testing (i.e., single pathogenic allele detected in autosomal recessive IRD) [6,21].

The cost and resolve rate of the various genetic testing modalities utilised in this study are outlined in Table 3. In total, 44% (*n* = 9) and 25% (*n* = 4) of re-tested patients had a genetically diagnostic result from single gene testing and WES, respectively. Thus, the most cost-effective initial genetic testing approach remains pNGS with the use of more costly and bioinformatic resource-consuming modalities reserved for genetically unresolved cases applied after reassessment of phenotype. The relatively low resolve rate for single gene sequencing may represent spurious 1st-tier variant detection, skewing focus to a single gene when a broader re-screening approach (e.g., WGS, array comparative genomic hybridisation) may yield an increased likelihood of resolution [22,23]. Closer scrutiny of single variants detected by pNGS (such as ACMG class and in silico functional analysis) will be undertaken prior to committing to 2nd-tier testing choice. A low resolve rate with WES may represent the majority of exonic variants being already screened with pNGS; in this case, WGS may be a more suitable approach to interrogate the deep intronic variants or structural variants of genes of interest. Primary application of these broader techniques to an untested IRD population may resolve significantly higher proportions of cases but at greater relative cost than pNGS [21,24].

Ongoing revision and upgrading of the NGS panel design allows the reapplication of this technique to existing DNA samples, which is the most cost-effective approach, as the primary cost of pNGS is preparation, sequencing, and panel design [25]. The total cost incurred to reassess the 34 patients was €23,200 (mean €1450 per resolved case) on top of the original pNGS costs and an additional €10,600 for estimated cost of clinicians and genetic counsellors).

### 4.3. Examples of Resolution Problems and Their Solutions

In research sequencing endeavors, there is a need to maximise the number of patients that can be analysed on a single sequencing run in order to optimise cost effectiveness. This aggressive pooling approach during sample preparation can result in some patients falling below the acceptable sequencing coverage thresholds. In certain cases, a specific phenotype may allow for the manual review of plausible candidate genes in the sequencing data (i.e., BAM files). This was the scenario for pedigree #23. Re-phenotyping of patient #170 revealed additional systemic features including diabetes mellitus and polydactyly in addition to teenage-onset RP in keeping with Bardet–Biedl syndrome (BBS, OMIM#209900). This narrowed the scope of the genetic search from all genes associated with autosomal recessive RP (i.e., 30 non-syndromic and > 27 syndromic) [26,27] to BBS-associated genes (*n* = 16), of which two (i.e., *BBS1* and *BBS10*) explain 45% of BBS cases [28]. Despite sub-optimal sequencing coverage, this refined phenotype allowed for the targeted direct manual inspection of BBS-associated genes, revealing two clearly pathogenic frameshift variants in the *BBS10* gene: c.2119_2120del, p.(Val707*) and c.687del, p.(Val230Phefs*7). These variants were then confirmed by direct sequencing.

Panel-based sequencing typically relies heavily on amplification methods to efficiently capture gene panels and incorporate necessary components (e.g., indexing tags for multiplexing DNA) for parallel sequencing. As a result, regions of DNA that are difficult to amplify may sequence poorly. A prime example in the IRD gene panel is the ORF15 region of the *RPGR* gene (OMIM*312610), which includes highly repetitive regions with sequences of low nucleotide diversity (e.g., purine-rich). Cloning allowing bidirectional sequence reads of such regions has been demonstrated as a cost-effective way to overcome this issue [29,30]. The *RPGR* ORF15 region is a crucial locus to assess X-linked RP (XLRP) pedigrees, as ≈60% of variants causing XLRP may be found in this mutational ‘hot spot’ [31]. Pedigrees that were resolved by the resequencing of *RPGR* include pedigree #26 (resolved by an expanded pNGS (351 genes) at an accredited laboratory) and pedigree #27 (resolved using trio WES) [32]. In retrospect, the variant identified in pedigree #27 was identifiable with the expanded pNGS (351 genes, commercial laboratory); their techniques to achieve better amplification/coverage of the ORF15 region are proprietary (commercial laboratory) but likely reflect superior capture or hybridisation methods than those used in the research laboratory and lends further credence to pNGS use as 1st-tier investigation for IRDs.

In rare circumstances, a patient’s zygosity of a variant may be misdetermined by alignment methods. This can result in a patient appearing to have a single variant in a gene associated with a recessive mode of inheritance, when in actuality, the patient is homozygous for that variant. In the case of pedigree #19 (*CFAP410* OMIM*603191), this was attributed to index hopping in the sequencing run, which is a sequencing phenomenon known to occur when multiplexing samples [33]. It occurs when the sequencing platform erroneously attempts to demultiplex the pool of samples and misassigns sequencing reads to the wrong sample of origin. In the case of pedigree #19, this was sufficient to result in a homozygous variant to be classified as heterozygous.

Panel-based sequencing frequently places a primary focus on exonic DNA regions. Canonical splice site variants can be routinely detected by any exon-focused approach given proximity to the exon, while the opposite is true for most deep intronic variants. However, variants in close proximity to exons may be inconsistently detected by exon-focused methods. The detection of variants located near exons depends on several variables such as capture probe design, efficiency of probe hybridisation, sequence coverage depth, as well as the browser extensible data file used for analysis. To ensure consistency and to simplify the variant interpretation process, it is possible to establish bioinformatic cut-offs based on purely exonic variants and canonical splice variants (±1–2 nucleotides from the exon). For this reason, some pathogenic intronic variants may be filtered out during the variant filtering process. Additionally, sequence coverage depth may significantly drop off outside of the exon target, resulting in near-exonic variants falling below detection thresholds: for example, near-exon aberrant RNA (NEAR) variants and deep-intronic variants [34]. A relevant example here is pedigree #22, whom had one variant identified on the initial pNGS run (*ABCA4* c.4363T > C, p.Cys1455Arg), and through retesting (single gene sequencing of the *ABCA4* gene), a second pathogenic variant in *ABCA4* c.4253 + 43G > A, p.Ile1377Hisfs*3 has been identified.

For clinically recessive cases, the initial detection of one likely causative allele in an appropriate IRD-associated gene adds extra evidence supportive of a clinical/genetic diagnosis of IRD. Single gene testing is appropriate in this situation, and 2nd-tier testing of intronic regions within the candidate gene identified a second pathogenic variant in 44% (*n* = 4/9) of the cohort tested in this manner (Table 2). Deep intronic variants resulting in activation of cryptic splice sites account for up to 5% of all known pathogenic *ABCA4* alleles [35,36], and in our cohort, both cases identified a second variant in the *ABCA4* gene (pedigree #21 novel non-canonical splice site variant/NEAR; pedigree #22 NEAR intronic variant). For 56% of partially resolved cases that were not resolved by 2nd-tier single gene sequencing, perhaps the originally 1st-tier pNGS-detected heterozygous variants (single alleles) were spurious, leading to further assessment of an unrelated gene. In fact, the single alleles detected by 1st-tier pNGS were ACMG class 5 (i.e., pathogenic) for pedigrees resolved by 2nd-tier single gene sequencing, while the five unresolved pedigrees were class 3 (variant of unknown significance) or 4 (likely pathogenic). In light of this, greater caution will be taken prior to this 2nd-tier approach, with only single variants with ACMG class 5 progressing to single gene sequencing and rigorous in silico analyses performed on < class 5 or novel variants prior to a decision regarding 2nd-tier genetic testing modalities (i.e., further evaluate the gene in question or reassess a wider range of IRD-implicated genes). If the initial variant cannot be upgraded to ACMG class 5, then broader genetic screening techniques, such as WES or WGS, are more appropriate. Although WGS is a powerful technique for resolving such issues, currently, the resource demand (financial and infrastructure) is prohibitive, as 1st- or 2nd-tier screening unless other more affordable yet highly effective options have been exhausted.

WES is gaining popularity as a primary method for the genetic assessment for IRDs [31]. Benefits include the ability to store entire exomes and apply a ‘virtual panel’ for genes of interest in specific areas of medicine [37]. This allows for the expansion of gene panels for subsequent re-analysis in cases where new genotype–phenotype associations have been established in the time since the original interpretation without the need for resequencing. Alternate gene panels (e.g., for other body systems, e.g., cardiac, neurological) can be applied to the pre-existing WES data at a later date if new phenotypic detail becomes apparent as well as for use in establishing population-specific variant frequency. Costs are upfront in terms of DNA sequencing, and data can be securely stored for future research or clinical applications. Published series report a 49–63% genetic resolve rate for their IRD cohorts using WES [22,31,38,39]; thus, the financial and efficiency benefit remains with pNGS as in our pathway [4,5]. Although a panel-based method eliminates the possibility of re-analysis with a broader virtual gene panel, this is less likely to be an issue for conditions whereby the gene associations have been largely already established. For IRDs, this is evident from the relatively small number of new gene associations in recent years [1].

Approximately 70–80% of IRDs are resolved by analysis of exons (e.g., by pNGS), while deep intronic mutations may account for 1.4–25%, and copy number variations account for up to 9% of IRDs and are unlikely to be detected by short read sequencing such as NGS [5,35,40,41]. The use of WGS as a primary genetic investigation is not practical at this time due to the high costs, large bioinformatic workload, and limited addition of genetic resolve rate over 1st-tier pNGS [30]. This broader sequencing approach may obscure the true genotype by detecting rare non-pathogenic polymorphisms, causing delay in reaching an accurate harmonised molecular/clinical diagnosis. As per WES, digital storage of WGS data from unresolved cases/pedigrees may allow virtual panel tests to be performed in the future and for new gene associations to be made. The added risk in 1st-tier WGS approaches is the workload required to process the output data, which may require artificial intelligence algorithms to create actionable clinical outcomes within a reasonable timeframe [42], and detection of significant secondary findings relevant to other body systems (e.g., cancer risk genes) [10,43]. However, recent data from the UK 100,000 genomes project showed >40% genetic diagnosis rate using 2nd-tier WGS for heritable ophthalmic disease with previously negative 1st-tier genetic testing (e.g., pNGS, WES); WGS addressed issues including non-coding/structural variants, mitochondrial DNA variants, and poor coverage by exon-based approaches [8,44].

The ‘cost-effectiveness’ of genetic testing for IRDs includes not only the direct costs of genetic sequencing but also the surrounding infrastructure (e.g., genetic counsellor, MDT meeting, molecular geneticist/scientist time/training, clinic use). The broader the scope of a test (e.g., WGS), the greater the yield of non-diagnostic (potentially misleading) findings, including secondary findings, which must be reported [10,16,43]. The expenditure of limited resources on the use and interpretation of 1st-tier WGS in a health service of limited size reduces the number of patients/pedigrees that can be served. Thus, pNGS maximises the genetic/diagnostic resolution rate for IRD in small/medium-sized countries with the reservation of broader tests for the minority of unresolved cases. Large collaborative groups involved in such multicenter WGS projects will likely identify condition-specific algorithms for optimal genetic resolution yield while having strategies for addressing secondary findings from these broad-scope techniques. WGS may supersede pNGS as 1st-tier sequencing for IRDs on a global scale once further cost reduction and better interpretation/infrastructure guidelines are in place.

### 4.4. Relevance to Gene Therapies

The need to achieve an accurate genetic diagnosis for IRD patients is increasingly important as novel gene therapies are in clinical trials for a growing number of etiologies (Table 4). Few (15.6% (*n* = 5)) of the patients in this study are eligible for upcoming gene therapy clinical trials for *RPGR* gene mutations (NCT03316560, NCT03252847, and NCT03116113) for which specific genotype parameters are an inclusion criteria prerequisite. A validated genetic diagnosis also allows accurate genetic counselling and family planning, as many patients with IRD are in the reproductive age (e.g., prenatal diagnosis and pre-implantation genetic diagnosis) [2,3].

### 4.5. Limitations

This study was carried out during the global SARS-CoV-2 pandemic, and thus, some patients were unwilling to travel for further in-person clinical assessment, potentially reducing the total power of the investigations in terms of final genetic resolution rate. Two patients were deceased by time of reassessment, and thus, further genetic testing was not carried out. Trio WES was not possible on five patients, as relevant family members were not available (e.g., deceased, out of country, unwilling to attend/participate during SARS-CoV-2 pandemic). IRDs are rare by definition; thus, despite being a national effort including over 1000 patients, statistical analyses to make definitive conclusions regarding the most appropriate genetic testing methodology for each gene/variant cannot be reliably drawn from the data included herein. The most powerful evidence for initial pNGS is the cost savings accrued and high genetic resolve rate, which then frees funding for more expensive 2nd-tier genomic techniques to be used in a targeted manner, as outlined above once the phenotype has been refined.

## 5. Conclusions

This study showed that 2nd-tier genetic testing resolved 56% of previously unresolved pedigrees (pathogenic variants in IRD-implicated genes), leading to an overall resolve rate of 92% (388/423). Second-tier genetic testing should be guided by detailed clinical (i.e., reassessment, multimodal imaging, electrophysiology) and genetic (i.e., single alleles in AR disease) indications to achieve a molecular diagnosis of IRD in the most cost-effective manner. Detailed pedigree phenotyping can help to reclassify disease and guide non-IRD patients (e.g., AMD, uveitis) into more appropriate care pathways while preventing further unnecessary genetic testing. The glass ceiling of sequencing resolution in international data may represent a patient population including a proportion of non-inherited pathology. Diagnostic refinement may improve genetic resolution rates for true cases of IRD.

## Figures and Tables

**Figure 1 ijms-23-00995-f001:**
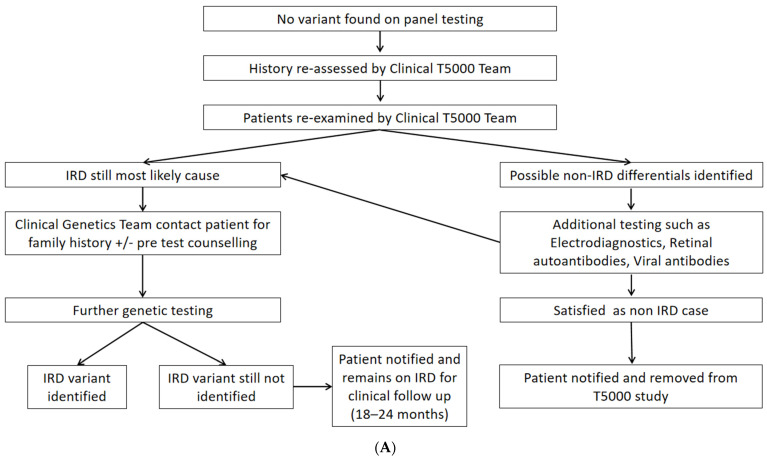
Proposed algorithm for clinical and genetic reassessment of ‘gene-negative’ cases. (**A**) Algorithm for clinical reassessment of ‘gene-negative’ cases. (**B**) Algorithm for selecting the most appropriate further genetic testing modalities. * https://blueprintgenetics.com/tests/panels/ophthalmology/retinal-dystrophy-panel/ (accessed on 8 November 2021).

**Figure 2 ijms-23-00995-f002:**
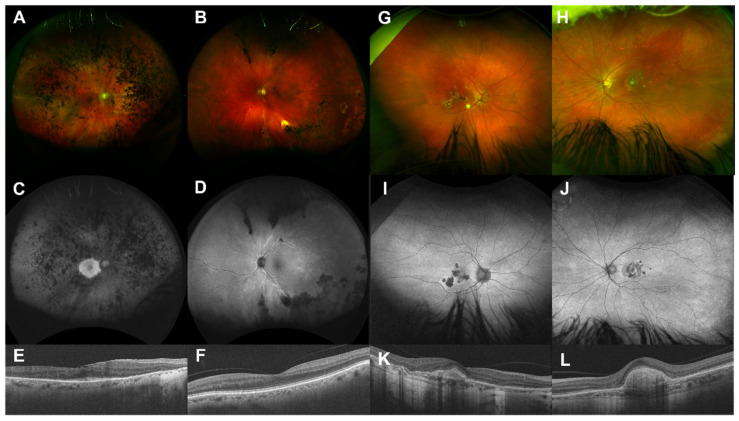
Non-IRD (‘gene negative’) examples with markedly asymmetric disease. Pedigree #2: Colour fundus photographs a 70-year-old man with markedly asymmetrical retinal pigmentation. The right eye (**A**) has more classic features of retinitis pigmentosa (RP) while the left eye (**B**) has less pronounced paraarteriolar intraretinal pigment migration only. Fundus autofluorescence delineates a small central island of residual RPE in the right eye (**C**) while the left eye (**D**) shows subtle hyperautofluorescent paravascular changes not in keeping with RP. Optical coherence tomography confirms asymmetric disease with relative sparing of the central macular outer retina in the right eye (**E**) with entirely normal retinal lamination in the left eye (**F**). No relevant genetic variants were detected on pNGS. This patient had a history of childhood meningitis with no family history, stable visual acuity (6/12 right, 6/6 left), and no progression of visual symptoms. Clinical consensus opinion was reached of asymmetric post-inflammatory pigmentary retinal changes, and no further genetic testing was indicated. Pedigree #3: Colour fundus photographs (**G**,**H**) of a 47-year-old woman with asymmetrical macular atrophy. Autofluorescence (**I**,**J**) shows hypoautofluorescent areas of macular atrophy with surrounding/intervening areas of hyperautofluorescence. Retinal vasculature and periphery are otherwise normal. OCT (**K**,**L**) shows outer retinal atrophy, subretinal fibrosis, and focal choroidal attenuation. The clinical phenotype was reassessed, and a diagnosis of punctate inner choroidopathy was made, with the decision to not pursue further genetic testing. The patient was referred to the uveitis clinic.

**Figure 3 ijms-23-00995-f003:**
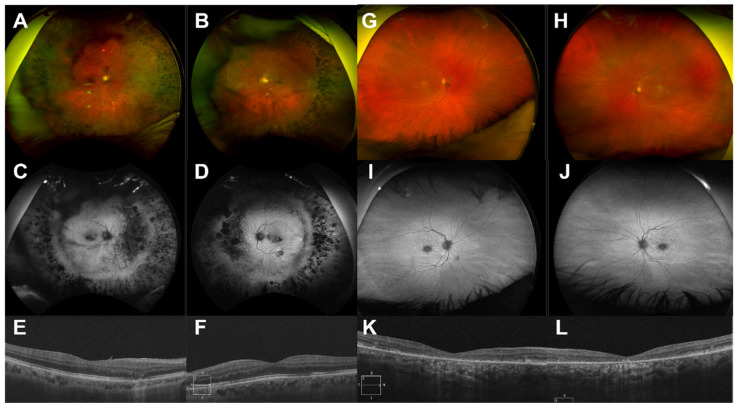
Multimodal imaging (MMI) of cases consistent with IRD. Pedigree #18: (**A**,**B**): Colour fundus photographs demonstrating symmetrical predominantly midperipheral pigmentary changes. (**C**,**D**): Autofluorescence showing patchy midperipheral and focal posterior pole hypoautofluorescence consistent with retinal pigment epithelium (RPE) atrophy. (**E**,**F**): OCT demonstrating predominant preservation of photoreceptor/RPE laminae with a focal nasal defect of photoreceptor inner segments (IS) in E and multiple (nasal and temporal) IS/RPE defects in F. A second pathogenic *PEX7* variant (OMIM*601757, c.40A > C, p.Thr14Pro) was detected via single gene sequencing, confirming a genetic diagnosis of autosomal recessive (AR) Refsum disease (OMIM#614879). This patient also had systemic features of this condition including ataxia. Pedigree #22: (**G**,**H**): Colour fundus photographs showing macular atrophy with surrounding subretinal flecks with sparing outside the vascular arcades. (**I**,**J**): Autofluorescence confirms foveal hypoautofluorescence surrounded by hyperautofluorescent flecks mostly confined to the macula with some flecks nasal to the optic nerve head in J. (**K**,**L**): OCT showing foveal outer retinal atrophy. These multimodal imaging features are in keeping with AR Stargardt disease (OMIM#248200). Single gene (i.e., *ABCA4*) testing allowed the detection of a second pathogenic allele (c.4253 + 43G > A p.[=, Ile1377Hisfs*3]), genetically resolving this case.

**Table 1 ijms-23-00995-t001:** Non-IRD diagnosis categories and demographics.

Group	Mean Age, Years (SD)	Female (%)	*n* = (%)
Posterior Uveitis	51 (16.39)	57%	7 (39%)
AMD	76 (8.39)	25%	4 (22%)
Myopic/Pachychoroid Degeneration	45.5 (3.54)	50%	2 (11%)
ION	50	100%	1 (6%)
Normal	49 (12.92)	75%	4 (22%)

AMD—age-related macular degeneration. ION—inherited optic neuropathy. SD—standard deviation.

**Table 2 ijms-23-00995-t002:** Summary of results obtained after additional genetic testing. Pedigrees 18–27 are resolved, and pedigrees 28–36 remained unresolved after additional testing. Variants in plain text were identified from the primary pNGS run, and variants in **bold** were detected with further testing modalities as outlined in the ‘Method’ column. All variants reported here were ACMG class 5 (pathogenic).

Pedigree	N =	Phenotype	Inheritance	Gene	Variant 1	Variant 2	Method	1st-Tier Problem
1–17	18	Non-IRD	-	-	-	-	-	-
18	1	Refsum Disease	AR	*PEX7*	c.875T > A, p.Leu292 *	**c.40A > C, p.Thr14Pro**	Single gene testing	Limited coverage
19	2	EOSRD	AR	*CFAP410*	c.218G > C, p.Arg73Pro	**c.218G > C, p.Arg73Pro**	Repeat pNGS (R)	Misaligned reads, index hopping
20	1	sRP	AR	*FLVCR1*	**c.1022A > G, p.Tyr341Cys**	**c.1307 + 5G > T** †	Repeat pNGS (R)	Additional phenotype information
21	1	STGD	AR	*ABCA4*	c.752del, p.Phe251Serfs*11 †	**c.5461 − 10T > C** **p.Thr1821Aspfs*6, Thr1821Valfs*13**	Single gene testing	Intronic variant
22	1	STGD	AR	*ABCA4*	c.4363T > C, p.Cys1455Arg	**c.4253 + 43G > A** **p.Ile1377Hisfs*3**	Single gene testing	Intronic variant
23	1	BBS	AR	*BBS10*	c.2119_2120del, p.Val707 *	**c.687del, p.Val230Phefs*7**	Repeat pNGS (R)	Poor coverage
24	3	nsRP	AD	*RP1*	**c.2321_2322insAlu**	-	Repeat pNGS (A)	Complex structural variant
25	1	nsRP	AR	*EYS*	c.2620C > T, p.Gln874 *	**c.(?_-538-1) _(2023+1_2024-1)del** †	Single gene testing	Copy number variants
26	3	nsRP	XL	*RPGR*	**c.2777_2778del, p.Glu926Glyfs*152** †	-	Repeat pNGS (A)	Low complexity ORF15 region
27	2	nsRP	XL	*RPGR*	**c.2571_2572del, p.Glu859Glyfs*219**	-	Trio WES	Low complexity ORF15 region
28	1	sMD	AR	*CNNM4*	c.1660G > T, p.Ala554Ser	Unresolved	Single gene testing	-
29	1	USH	AR	*CDH23*	c.289-1G > A, p.Arg964Gln	Unresolved	Single gene testing	-
30	1	nsRP	AR	*TRIM32*	c.691del, p.Ala231Glnfs*21	Unresolved	Single gene testing	-
31	1	USH	AR	*ADGRV1*	c.18025C > T, p.Arg6009 *	Unresolved	Single gene testing	-
32	1	BBS	AR	*BBS1*	c.478C > T, p.Arg160Trp	Unresolved	Single gene testing	-
33	3	nsRP	AD *	-	Unresolved	-	Trio WES	-
34	1	nsRP	AR *	-	Unresolved	Unresolved	Trio WES	-
35	9	VRO	AD *	-	Unresolved	-	Trio WES	-
36–52	16	Still consistent with IRDs	-	-	Unresolved	Unresolved	-	Retest delayed due to SARS-CoV-2 pandemic

AD—Autosomal Dominant. AR—Autosomal Recessive. XL—X-linked. BBS—Bardet–Biedl Syndrome. EOSRD—Early-onset severe retinal dystrophy. sRP—Syndromic Retinitis Pigmentosa. nsRP—Non-syndromic Retinitis Pigmentosa. sMD—Syndromic Macular Dystrophy. STGD—Stargardt Disease. USH—Usher Syndrome. VRO—Vitreoretinopathy. * Presumed inheritance pattern based on available family history. (A)—repeat pNGS at accredited laboratory. (R)—repeat pNGS at research laboratory. † novel variant.

**Table 3 ijms-23-00995-t003:** Indicative total cost of genetic testing approaches used in this study (Trinity College Dublin, Ireland and Blueprint Genetics, Finland) and resolve rate for each testing modality. pNGS—panel-based Next-Generation Sequencing. WES—Whole Exome Sequencing.

Test Type	Cost	N = Per Test Type	Total Cost	Resolve Rate, % (*n*=)
pNGS (resolved at research laboratory + accredited laboratory validation)	€600 (€250 + €350)	441	€240,450 *	84.4% (372/441)
pNGS (negative research laboratory result + expanded accredited laboratory panel + validation at an accredited laboratory for other affected family members)	€1120 (€250 + €870 ± €350)	5 + 3 variant confirmation	€6350	100% (5/5)
Single Gene Testing (negative research laboratory result + accredited single gene test + validation at an accredited laboratory for other affected family members)	€700 (€250 + €450 ± €350)	9	€6300	44% (4/9)
WES/trio WES (negative research laboratory result + accredited trio WES + validation at an accredited laboratory for other affected family members)	€2550 (€250 + €2300 ± €350)	4 + 1 variant confirmation	€10,550	25% (1/4) **

* Being unresolved, the 69 unresolved cases discussed in this paper only underwent initial research-grade pNGS prior to this study, and thus, this total represents (372 × €600) + (69 × €250). ** The case identified with WES could be solved using expanded pNGS (351 genes).

**Table 4 ijms-23-00995-t004:** List of ongoing IRD gene therapy clinical trials.

Gene	NCT Number	Technique	Phase	Status
**Rod-Cone Dystrophies**				
*MERTK*	NCT01482195	AAV	1/2	Completed
*PDE6B*	NCT03328130	AAV	1/2	Recruiting
*RHO*	NCT04123626	AON	1/2	Recruiting
*RPGR*	NCT03252847	AAV	1/2	Completed
	NCT03116113	AAV	1/2	Completed
	NCT03316560	AAV	1/2	Recruiting
*RLBP1*	NCT03374657	AAV	1/2	Recruiting
*USH2A*	NCT03780257	AON	1/2	Not recruiting
*MYO7A*	NCT01505062	LV	1/2	Terminated
**Macular/Cone Dystrophies or Cone Dysfunction Syndromes**				
*RS1*	NCT02416622	AAV	1/2	Terminated
	NCT02317887	AAV	1/2	Recruiting
*ABCA4*	NCT01367444	LV	1/2	Terminated
*CNGB3*	NCT03001310	AAV	1/2	Completed
	NCT02599922	AAV	1/2	Recruiting
*CNGA3*	NCT03758404	AAV	1/2	Completed
	NCT02935517	AAV	1/2	Recruiting
	NCT02610582	AAV	1/2	Recruiting
**Leber Congenital Amaurosis**				
*RPE65*	NCT02781480 NCT02946879	AAV	1/2	Recruiting
	NCT00643747	AAV	1/2	Completed
	NCT01496040	AAV	1/2	Completed
	NCT00821340	AAV	1	Completed
	NCT00749957	AAV	1/2	Completed
	NCT00481546	AAV	1	Completed
*GUCY2D*	NCT03920007	AAV	1/2	Recruiting
*CEP290*	NCT03913143	AON	3	Not recruiting
	NCT03872479	Gene editing	1/2	Recruiting
**Choroidal Dystrophies**				
*CHM*	NCT02341807	AAV	1/2	Completed
	NCT02671539	AAV	2	Completed
	NCT01461213	AAV	1/2	Completed
	NCT02077361	AAV	1/2	Not recruiting
	NCT02553135	AAV	1/2	Completed
	NCT03507686 NCT03496012	AAV	2 3	Completed Completed
	NCT02407678	AAV	2	Completed
	NCT04483440	AAV	1	Recruiting

AAV = adeno-associated virus. AON = antisense oligonucleotide. LV = lentiviral vector NCT = Reference number for study on clinicaltrials.gov.

## Data Availability

Anonymised source data are available upon reasonable request.

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
