# Peer review of "Clinical and Genetic Re-Evaluation of Inherited Retinal Degeneration Pedigrees following Initial Negative Findings on Panel-Based Next Generation Sequencing"

_ijms, 2022, doi:10.3390/ijms23020995_

Round 1
Reviewer 1 Report
This is a well-written, comprehensive and interesting analysis of the process of reassessment of IRD patients with initial negative results from panel testing, and the rationale, outcomes and cost of second-tier genetic testing approaches. The authors also conclude that panel testing is the most cost effective first-tier approach for molecular assessment of diverse IRD populations.
With whole genome sequencing now available in the UK for example and the method of choice for inherited retinal diseases, this analysis of the cost-effectiveness of the different approaches is timely and important. However, I think that the authors’ approach of cost-effectiveness is also limiting and omits an analysis of what would be a fundamentally idealised approach. This would structure many of the comments made in the Discussion into a more critical appraisal of the options available.
A more long-term approach may consider that currently and more so over time, WGS is becoming significantly reduced in cost, the analysis pipeline more efficient and more effective. The advantages of covering the intronic variants, structural rearrangements, new gene discovery, research etc. are significant and WGS will always be the ultimate pathway to solving the unsolved. As it becomes more streamlined, will it not become the first-tier method of choice in all but the most pathognomonic of cases? The argument seems to go back and forth especially on page 10, and perhaps could be clarified.
Minor comments:
Page 1, Lines 38-39
Australia England and Wales because they are the only countries to report this or because other countries (Western, developed etc) have a different main cause of visual loss in the working age group? Please clarify.
Page 2, line 45
WGS does not necessarily have to report secondary findings. WGS through the NHS is currently not reporting additional findings at this stage. The 100,000 Genomes Project was reporting additional findings in the second phase, which is perhaps why, for now the NHS is not.
Page 2, lines 50-51.
Specifically mention the need for a genetic diagnosis for reproductive options such as prenatal diagnosis and pre-implantation genetic diagnosis either at this point or later when you mention genetic counselling and patients of reproductive age (Page 10, lines 342-343).
Page 2, lines 69-70
Patients with a negative result had their records reviewed. Were the bracketed assessments done at this stage or were they reviewed? It is not clear. You do not for example mention electrophysiology as one of the initial assessments. Are you implying that it should be done initially or saved for extra phenotyping?
Page 3 Figure 1.
Is further genetic testing required (left hand side of figure) meant to mean the larger Blueprint panel of 322 genes? It would be good to see a list
Also “family member available for testing” – I assume you mean a parent as this leads to trio WES. This should be made clearer who is included in this additional family member. Preferable to have another affected relative or an unaffected parent? Etc.
It would be interesting to see what the extra genes are in the 322 panel.
How many patients were solved by the Blueprint panel and which genes from the Blueprint panel were found in the patients solved in this way?
Page 5, Table 2
Does Repeat panel testing mean the Blueprint panel of 322 genes. Does that mean that CFAP41, FLVCR1, BBS10 etc were not in the original panel of 250 genes? Please could you clarify
Did the authors consider WGS as tier 3 for families 28-32? If not, why not?
Were all the variants in the table likely pathogenic/pathogenic? What about VUSs and upgrading them etc?
Page 6, line 132
What do you mean by pNGS repeated testing. Do you mean a repeat of the 250 gene panel or the 320 gene panel? Needs clarifying.
Page 6, line 135
What about the 66% AR cases not solved by single gene testing? Was WGS tried on those? Perhaps the solve rate would have been higher?
Page 6, lines 156-160
Mention of hydroxychloroquine toxicity, syphilis as other conditions that mimic IRDs if you think relevant
Page 7, line 194
Bioinformatic demands. What about in a system that can manage these demands, like the NHS, and with decreasing costs, is WGS not going to become the first-tier method of choice? With the analysis of the right panel within WGS? A panel that can easily expand.
Page 8, lines 218-222. Where cost is not an issue, WGS has the highest yield. This is an important point.
Page 8, line 213
‘And’ should be ‘an’
Page 9, line 252
Pedigree 27 was resolved using trio WES. In Table 2, it says it was resolved using trio WGS. Which one was it? Also, you conclude that the variant was identifiable with the expanded NGS panel of 322 genes. Does that mean that RPGR-ORF15 is one of the extra genes in the larger panel?
Isn’t the problem with RPGR-ORF15 the highly repetitive regions, so how does the panel test help?
Page 9, lines 260-263.
It is not clear how this is picked up and how it is resolved.
Page 9, paragraph lines 264-280
This is also where WGS would find the intronic variants directly.
Page 10. Line 298.
This can be done directly by WGS
Page 10, Lines 310-314
There will still be a percentage of unsolved with the panel testing. On this point, an advantage of WGS is exactly this point i.e. in the discovery of new gene associations
Page 10, lines 318-319
Again, there are two issues. Which is the best most comprehensive method – WGS because it will cover the 25% intronic variants and the CNVs, structural rearrangements. And the second issue of cost where panel testing is cheaper
Page 10, paragraph beginning line 315
Seems like you are saying here that WGS is the preferable form of testing.
Page 10, lines 320-322
Is the solution to the rare non-pathogenic polymorphisms, to not do WGS?
Page 10, line 327
“The added risk…” I am not sure what point you are making here
Reviewer 2 Report
The manuscript presents the results of a second tier genetic testing for patients who did not reach a positive genetic diagnosis for IRD, in a major Irish program focused on obtaining the genetic diagnosis of 5000 patients (T5000). The paper is well written and very easy to follow and understand, and the Discussion is very well presented. As a whole, the article may be of interest to other researchers working on genetic diagnosis of rare diseases (not exclusively ocular diseases), since the kind of considerations the authors pour into the text are of significance. Most of the reports publish the results on large cohorts of patients, but hardly any publication considers what is the result of a second check on the genetic tests previously classified as negative or inconclusive. In this sense, this is an interesting report, and the flow chart in Figure 1 is very illustrative.
For instance, the authors claim that an accurate clinical diagnosis is relevant in order to focus on patients who might be affected of inherited retinal dystrophies (otherwise testing patients not affected by IRDs using a panel of IRD genes does not make any sense and the results will indeed be negative), and also on how to select the genes to be re-evaluated in order to find missing mutations. This type of considerations may seem obvious to a human geneticist, but indeed, it is an emphasis that may have an interest to a more general reader.
However, there are several points that clearly need to be addressed, particularly on the Methods (which is a very short briefing), and on the Results section. My concerns are as follows:
1) The article finally presents the positive genetic test of 10 pedigrees obtained on a second round. All of them could have been correctly tested on the first round if the gene panel had worked properly for some exons (poorly covered, for instance), or a better bioinformatics program had been applied in order not to miss mutations in the flanking exon-intron regions. Therefore, the whole point is to find how to address this missing information. How the second round was able to find the mutations? The authors do not explain, what improvement has been done on "Repeat Panel testing" or what does "Single gene testing" entail. Some reference is made on "targeted direct manual inspection", which should have been applied probably on the first genetic test round. The method the authors have used to identify this missing mutations should be presented clearly in the Methods and Results sections, so as to be more informative for other geneticist who have to deal with similar results.
2) In no place the authors say that identified mutations were confirmed by Sanger sequencing. Sanger sequencing allows to check whether the mutation identified is real or an artifact, and also allows to clearly identify if a patient is heterozygous or homozygous for a given mutation, lines 255-263.
3) Table 2, which shows the summary results, do not indicate whether the new mutations identified in patients were novel or previously reported. This information could be added easily, in the same Table or in Supplementary Material.
4) Geneticists working on IRD genetic testing know the frequency of intronic variants in ABCA4, or in CEP290, USH2A, CDH23... Some of them are really frequent among some cohorts. Target gene panels are customized, and many include already reported mutations so that a second deep intronic allele, which could go undetected otherwise is not missed. This is not presented neither discussed as a possible explanation for the unsolved cases.
5) Some of the results were obtained in a research laboratory, some in an accredited laboratory. It is not clear what the final message is. Is it best to improve the results of a panel made in a research laboratory, so as to increase the final yield? or just directly send the samples to be analysed in an accredited laboratory that provide a positive result? If the message is that it is best to perform a target gene panel, how to improve it so as to increase the percentage of positive results?
Round 2
Reviewer 2 Report
The authors have addressed my concerns and I have no further comment.